# Pharmacokinetic Study of Mucoadhesive Itopride Hydrochloride In Situ Nasal Gel Formulations in a Comparative In Vivo Study and Histopathological Safety Evaluation

**Maha A. Marzouk [1], Dina A. Osman [1], Amany I. Abd El-Fattah [1] and Reem A. Aldeeb [2,*]**

[1] Department of Pharmaceutics and Pharmaceutical Technology, Faculty of Pharmacy, Al-Azhar University, Cairo 11751, Egypt; mahamarzouk.pharmg@azhar.edu.eg (M.A.M.); dosman1971@gmail.com (D.A.O.); dr.amanyhallol@gmail.com (A.I.A.E.-F.)

[2] Department of Pharmaceutics, College of Pharmaceutical Sciences and Drug Manufacturing, Misr University for Science and Technology, Giza 12566, Egypt

\* Correspondence: reem.eldeeb@must.edu.eg; Tel.: +20-1001408552

**Abstract:** Hepatic first-pass metabolism has been a major cause of reduced bioavailability for many drugs. Using the nasal route as an alternative route to deliver drugs to the systemic circulation provided the solution to this problem. One of the drugs which are highly affected by first-pass metabolism is itopride hydrochloride (ITO HCl). It is a prokinetic agent used for the treatment of various gastrointestinal motility disorders, mainly gastroesophageal reflux. The objective of this study was to determine the pharmacokinetic parameters of selected mucoadhesive in situ nasal gel formulations (F1 and F17) of itopride hydrochloride (ITO HCl) and to evaluate their safety after topical application on the nasal mucosa. The tested formulations contained 18% *w/v* poloxamer 407 with 0.5% *w/v* of HPMC K4M (F1), or with 0.5% *w/v* MC (F17). A randomized cross-over study was done on six rabbits after administration of F1, F17, and commercial oral tablets (Ganaton®). Plasma levels were assessed using high-performance liquid chromatography (HPLC) to compare the nasal gel formulations with the conventional oral tablets. Histopathological study of the nasal mucosa was performed in rats after nasal application of both in situ gel formulas. The in vivo pharmacokinetic profiles of in situ nasal gel formulas F1 and F17 provided showed improvement in $C_{max}$, $K_e$, $t_{1/2}$, $AUC_{0-24}$, $AUC_{24-inf}$, $AUC_{0-inf}$, $AUMC_{24-inf}$, $AUMC_{0-inf}$, MRT, $V_d$, and $C_{max}/AUC_{0-24}$ values over commercial tablets ($p < 0.05$). No statistically significant difference was found between both nasal gel formulas (F1 and F17). The percentage relative bioavailability of ITO HCl nasal in situ gel F1 and F17 was found to be 171.22% and 178.91%, respectively, in comparison with the commercial tablet. Histopathological study of the nasal mucosa revealed the safety of nasal in situ gel formulations to the nasal mucosa after 14 days of application. The study showed that the formulation of itopride hydrochloride as a mucoadhesive in situ nasal gel has enhanced the drug bioavailability due to avoidance of first-pass metabolism. The study points to the potential of mucoadhesive nasal in situ gel in terms of safety and efficiency.

**Keywords:** itopride hydrochloride; nasal drug delivery; in situ gel; animal study; bioavailability; histopathology; in vivo

## 1. Introduction

The goal of any drug delivery system is to deliver the appropriate amount of therapeutic drug to the intended site in the body. For systemic circulation, GIT is the main route of drug administration. However, for some drugs, this route of administration possesses various problems. Thus, drugs administered through GIT are susceptible to acid hydrolysis and may undergo extensive first-pass metabolism. This may lead to poor bioavailability of the drug when administered via the oral route. To avoid this issue, alternative drug administration is required [1,2].

Nasal drug delivery becomes a more efficient and popular route not only for local effect but also for competing with the GIT for systemic drug administration. It provides an easy non-invasive way to apply medications, which allows for self-administration by removing the chance of unwanted painful conditions associated with the injection form of drug delivery. Further advantages for systemic drug delivery via the nasal route include providing a large surface area for drug absorption, avoidance of hepatic first-pass metabolism and drug degradation in the gastrointestinal tract, and relatively high bioavailability [3–5].

Mucoadhesive nasal in situ gel provides all the benefits of mucoadhesive drug delivery systems together with the advantages of nasal drug delivery systems. The relatively low residence time of the drug in the nasal cavity affects the bioavailability of the drug. The possible strategy to improve the residence time is to decrease rapid mucociliary clearance using mucoadhesive formulations [6,7].

ITO HCl is rapidly, and extensively absorbed, and peak serum concentrations are achieved within 35 min after oral dosing, thus it has a rapid onset of action with a short half-life ($t_{1/2} < 6$ h). Its relative bioavailability is calculated to be 60% due to the extensive liver first-pass metabolism. It is metabolized in the liver via N-oxidation to inactive metabolites and excreted mainly by the kidneys as metabolites and unchanged drugs [8].

Studying the bioavailability and pharmacokinetic parameters of dosage forms has become one of the ways to assess the in vivo performance following the development of drug formulations [9].

Mucoadhesive nasal delivery has the advantage of improving the residence time by decreasing rapid mucociliary clearance, and hence, increasing the bioavailability of nasal in situ gel, however equally important is their safety. The main area of concern is the local irritation, which may lead to damage or permanent change in the nasal mucosa. Examining tissues by light microscopy is a good indicator of the level of mucosal tissue inflammation [10].

Previous work was done for the formulation of ITO HCl thermoreversible mucoadhesive in situ nasal gel and found that 6 out of 26 formulations demonstrated good in vitro results. Only F1 and F17, which contain 18% *w/v* poloxamer 407 and 0.5% *w/v* of hydroxypropyl methylcellulose K4M or methylcellulose (MC), respectively, showed higher stability results as indicated by their higher $t_{90}$ values (days). The study recommended further in vivo investigations of both formulas [10].

This research aimed to determine the pharmacokinetic profiles of the prepared ITO HCl in situ nasal gel F1 and F17 and to compare it with the available commercial oral tablets (Ganaton®, ABBOTT, Chicago, IL, USA) in rabbit's plasma using the HPLC-UV detector method. Investigation of the efficacy of the prepared nasal gels was done through a relative bioavailability assessment. Histopathological study of the nasal mucosa after application of ITO HCl formulations was performed in rats.

## 2. Materials and Methods

### 2.1. Materials

Itopride hydrochloride (ITO HCl) and methylcellulose (MC) were kindly supplied by MUP Pharm for Pharmaceutical Industries (Nasr City, Egypt). Hydroxypropyl methylcellulose (HPMC K4M) and poloxamer 407 were kindly supplied by the Egyptian International Pharmaceutical Industries Co. (EPICO) (Ramadan City, Egypt). Benzalkonium chloride was obtained from EL-Gomhouria Co. (Cairo, Egypt). Disodium hydrogen phosphate, potassium dihydrogen phosphate, and sodium chloride were supplied from El-Nasr Pharmaceutical Co. (Cairo, Egypt). Ganaton® 50 mg tablet (ABBOTT, Chicago, IL, USA); levofloxacin; potassium dihydrogen phosphate; acetonitrile; O-phosphoric acid; dichloromethane; thiopental; and acetic acid were all purchased from their production companies.

*2.2. Methods*

2.2.1. Preparation of In Situ Nasal Gels

The selected in situ nasal gel formulations used in this study that were containing 5% *w/v* ITO HCl, 18% *w/v* P407, 0.9% *w/v* NaCl, 0.01% *w/v* benzalkonium chloride and mucoadhesive polymer were as follows: F1 (0.5% *w/v* HPMC K4 M) and F17 (0.5% *w/v* MC). In situ gels were prepared by a cold method described by Peedikayil S.S. and Vasantha V.P. (2015). ITO HCl, sodium chloride, benzalkonium chloride, and mucoadhesive polymer were dissolved in distilled water by agitation at room temperature. After cooling the solution to 4 °C (kept overnight to complete hydration of polymers), P407 was added slowly with stirring (thermostatically controlled magnetic stirrer, Clifton cerasti, Model C/STIR; UK). The resulting dispersion was then kept overnight at 4 °C until a clear transparent solution was formed and finally the volume was adjusted [1].

2.2.2. In Vivo Study and Evaluation of ITO HCl in Rabbit Plasma

The protocol for the in vivo evaluation of the prepared formulations was reviewed and approved by the ethical committee, Faculty of Pharmacy, Al-Azhar University. The experiments performed followed the regulations of the Guide for the Care and Use of Laboratory Animals [11].

2.2.3. Animal Handling, Study Design, and Drug Administration

Six male (*n* = 6) New Zealand white rabbits weighing 3.08 Kg ± 0.11 kg were used. They were individually kept in stainless steel cages and fed a commercial laboratory rabbit diet. The rabbits fasted for 12 h before and during the pharmacokinetic study with free access to water by *ad libitum*. The animals remained conscious throughout the duration of the experiments. A single-dose randomized cross-over study design with a wash out period of 7 days was followed (Table 1); rabbits received a dose of ITO HCl equivalent to 2.5 mg/kg. Group I administered the designed dose from the commercial tablets. A volume equivalent to 2.5 mg/kg ITO HCl from the developed in situ nasal gels F1 and F17 (0.15 mL equivalent to 7mg Itopride HCl) were deposited into the right nostril of group II and group III, respectively, using an insulin syringe [12,13].

**Table 1.** A schematic presentation and plan description for the in vivo cross-over study design.

| | Treatment period (I) | Wash Out Period (7 days) | Treatment period (II) | Wash Out Period (7 days) | Treatment period (III) |
|---|---|---|---|---|---|
| Screening Number = 6 | Group (I) *n* = 2 A, B | | Group (I) *n* = 2 E, F | | Group (I) *n* = 2 C, D |
| | tablets | | tablets | | tablets |
| | Group (II) *n* = 2 C, D | | Group (II) *n* = 2 A, B | | Group (II) *n* = 2 E, F |
| | F1 | | F1 | | F1 |
| | Group (III) *n* = 2 E, F | | Group (III) *n* = 2 C, D | | Group (III) *n* = 2A, B |
| | F17 | | F17 | | F17 |

2.2.4. Sample Collection

After administration of different formulations, a blood sample (1.5 mL) was withdrawn at different time intervals of 0.08, 0.25, 0.5, 0.75, 1, 2, 4, 6, 8, 12, and 24 h from the marginal ear vein of the rabbits. Blood samples were collected in EDTA tubes to avoid clotting and samples were centrifuged at 4000 rpm for 15 min to obtain the plasma. The separated plasma tubes were stored at −20 °C until assayed [14,15].

### 2.2.5. Chromatographic Procedure

The HPLC system (Agilent 1260) with a UV detector was used. All samples were assayed at ambient temperature using the HiQsil C18 column (25 cm), using Levofloxacin as an internal standard. A mixture of 75:25 *v/v* acetonitrile with 0.05 mM phosphate buffer solution was adjusted at pH 4.6 and used as a mobile phase. The mobile phase was filtered through a 0.45 µm membrane filter and was then degassed by ultrasonication (Elmasonic S30H, Elma; Germany) before usage. Analysis was run at a flow rate of 1 mL/min and the detection wavelength was set at $\lambda_{max}$ 258 nm [8].

### 2.2.6. Determination of ITO HCl in Rabbit Plasma

A volume of 225 µL of rabbit plasma was withdrawn, 25 µL of a standard solution of Levofloxacin was added as an internal standard (IS) to reach 250 µL as a final volume. Between each step, the solution was mixed well, 4ml of dichloromethane was added, vortexed (VWR VV3 S540 International West Charter; USA) well for 3 min, then centrifuge for 5 min at 5 °C at 5000 rpm. The supernatant was decanted and placed in a vacuum oven (VACUCELL VUS-B2V-M/VU 22, MMM Group; Germany) till complete evaporation. After drying, the residue was reconstituted with 200 µL of mobile phase, then injected into HPLC Agilent 1260 for analysis [8,16].

### 2.2.7. Histopathological Study

Six male albino rats, weighted 250–300 g, were equally divided into 2 groups. They were sedated with an intraperitoneal injection of thiopental (45 mg/kg) before each dosing to facilitate nasal administration. Each rat received a once-daily nasal administration of 20 µL of mucoadhesive in situ gel, either F1 or F17. The gel was applied to the right nostril for 14 days to test the effect of the gel on the nasal mucosa after long-time application, while the left nostril was left for control. After this period, the rats were sacrificed. The nasal mucosa with the epithelial cell membrane on each side was carefully separated from the bone. Nasal mucosa specimens were fixed in 10% neutral buffered formalin. The fixed specimens were then trimmed, washed, and dehydrated in ascending grades of alcohol, cleared in xylene, embedded in paraffin, sectioned at 4–6 µm thickness (transverse section T.S), and stained by hematoxylin and eosin (H&E) [17]. The slides of control and treated nasal mucosal tissues were examined using a light microscope (120 V model (Japan), equipped with Nikon camera 3200) [18,19].

For all animal studies, the experimental procedures were reviewed and approved by the ethical committee, Faculty of Pharmacy, Al-Azhar University.

## 3. Results and Discussion

### 3.1. In Vivo Study and Evaluation of ITO HCl in Rabbit Plasma

The rabbit's plasma concentrations of ITO HCl in the 3 studied groups (commercial oral tablets, F1, and F17) were assessed by HPLC and the pharmacokinetic parameters and relative bioavailability were calculated. Data were collected and statistically analyzed using Statistical Package for Social Sciences (SPSS/version 24) software [20]. Comparison between more than two populations was analyzed using the F-test (ANOVA), followed by a post hoc test (Tukey-Kramer multiple comparison test) to compare between every two groups [6]. The significance of data difference was conducted using an unpaired *t*-test (two-tailed). The obtained results were judged at a confidence level of $p < 0.05$. Table 2 shows the Itopride HCl plasma concentration (ng/mL) at specified time intervals presented as mean $\pm$ standard deviation (SD) in the three groups. There was a statistically significant difference between the commercial tablet and nasal gel (F1) at all time intervals (P1 < 0.05) except after 1 h and 4 h (P1 > 0.05). A significant difference in drug plasma concentration was found between the commercial tablet and nasal gel (F17) at all time intervals (P2 < 0.05) except after 0.25 h and 1 h (P2 > 0.05). On the other hand, there was no statistically significant difference between nasal gel (F1) and nasal gel (F17) (P3 > 0.05) at 0.08, 0.75, 1, 2, 12, and 24 h, while a significant difference in plasma concentration was found between

nasal gel (F1) and nasal gel (F17) (P3 < 0.05) at other time intervals (0.25, 0.5, 4, 6, and 8 h). This may be related to the difference in dissolution profiles of the formulas [10]. Mean plasma concentration-time profiles of oral and nasal administration of ITO HCl are shown in Figure 1.

**Table 2.** Itopride HCl mean plasma concentrations ± SD (ng/mL) at specified time intervals after administration of commercial oral tablets and in situ nasal gel F1 and F17 of ITO HCl in rabbits.

| Time (h) | Gp I (Commercial Tablet) | Gp II (Nasal Gel F1) | Gp III (Nasal Gel F17) | ANOVA *p* Value | P1 | P2 | P3 |
|---|---|---|---|---|---|---|---|
| 0.08 | 152.71 ± 12.57 | 224.96 ± 11.75 | 217.09 ± 10.33 | 70.216, 0.001 * | 0.001 * | 0.001 * | 0.258 |
| 0.25 | 244.37 ± 15.02 | 284.10 ± 9.29 | 252.40 ± 16.89 | 13.309, 0.003 * | 0.002 * | 0.340 | 0.01 * |
| 0.5 | 475.87 ± 26.89 | 397.28 ± 19.62 | 366.42 ± 13.86 | 44.093, 0.001 * | 0.011 * | 0.001 * | 0.021 * |
| 0.75 | 573.18 ± 53.32 | 473.47 ± 24.83 | 457.07 ± 15.39 | 19.226, 0.002 * | 0.013 * | 0.012 * | 0.431 |
| 1 | 514.65 ± 81.54 | 581.68 ± 21.70 | 546.59 ± 23.00 | 2.646, 0.104 | 0.063 | 0.290 | 0.247 |
| 2 | 247.60 ± 12.92 | 673.87 ± 18.86 | 664.79 ± 14.72 | 444.1, 0.001 * | 0.001 * | 0.001 * | 0.33 |
| 4 | 105.45 ± 10.94 | 113.94 ± 12.29 | 137.33 ± 10.63 | 12.790, 0.016 * | 0.213 | 0.001 * | 0.003 * |
| 6 | 47.46 ± 8.24 | 79.97 ± 6.29 | 91.02 ± 8.69 | 50.464, 0.001 * | 0.001 * | 0.002 * | 0.027 * |
| 8 | 25.54 ± 5.49 | 54.19 ± 9.73 | 74.35 ± 6.09 | 66.897, 0.001 * | 0.001 * | 0.0013 * | 0.0011 * |
| 12 | 15.82 ± 4.25 | 42.06 ± 5.20 | 45.11 ± 6.17 | 56.116, 0.001 * | 0.001 * | 0.001 * | 0.323 |
| 24 | 6.80 ± 2.03 | 23.44 ± 2.92 | 25.47 ± 4.02 | 65.560, 0.001 * | 0.001 * | 0.001 * | 0.275 |

P1 comparison between commercial tablet and nasal gel (F1); P2 comparison between commercial tablet and nasal gel (F17); P3 comparison between nasal gel (F1) and nasal gel (F17); * Significant difference.

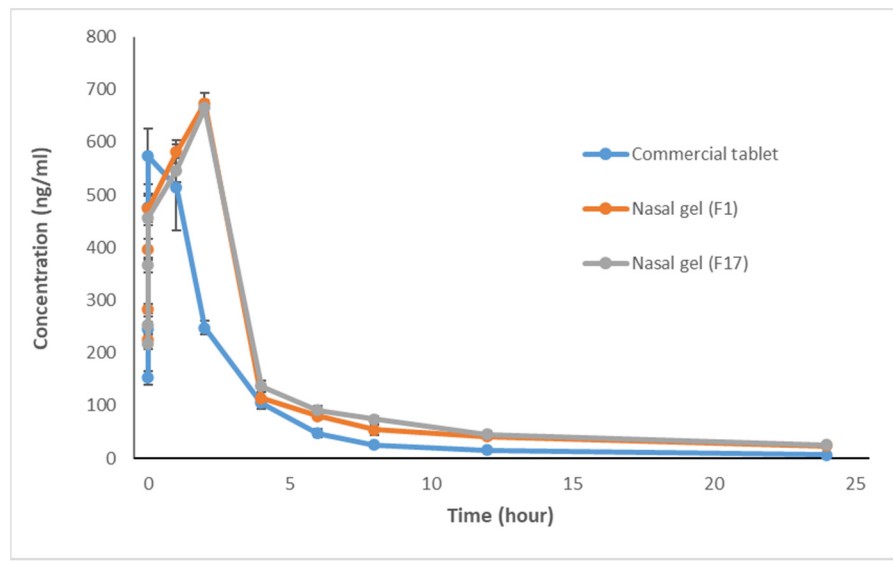

**Figure 1.** Mean plasma concentration-time profiles ± SD following administration of commercial oral tablets and in situ nasal gel formulations F1and F17 of ITO HCl in rabbits.

It was noticed that the commercial ITO HCl oral tablet has a mean peak concentration of 608.16 ng/mL and the time required to reach the peak was 0.75 to 1 h. On the other hand, in situ nasal gel formulations, F1, and F17 have mean peak concentrations of 673.87 and 664.79 ng/mL, respectively, with a time of 2 h to reach the peaks.

The maximum plasma concentration ($C_{max}$, ng/mL) and the time to reach this concentration ($T_{max}$, h) were determined from the data of plasma levels and were presented as mean ± standard deviation (SD). The trapezoidal rule was used to calculate the area under the curve from time 0 to 24 h ($AUC_{0-24}$, ng·h/mL). The area under the curve from time 0 h

to infinity (AUC$_{0-\infty}$, ng·h/mL), elimination rate constant (Ke, h$^{-1}$), half-life (t$_{1/2}$, h), and volume of distribution (V$_d$, L) were all calculated using the suitable equations [21]. The pharmacokinetic profiles are illustrated in Table 3.

**Table 3.** Mean pharmacokinetic parameters (±SD) after oral administration of commercial tablets and nasal administration of in situ gel formulations F1 and F17 to rabbits.

| Parameter | Commercial Tablet | Nasal Gel F1 | Nasal Gel F17 |
|---|---|---|---|
| C$_{max}$ (ng/mL) | 608.16 ± 28.10 | 673.87 ± 18.86 | 664.79 ± 14.72 |
| T$_{max}$ (h) | 0.83 ± 0.13 | 2 ± 0 | 2 ± 0 |
| t$_{1/2el}$ (h) | 4.23 ± 0.47 | 9.61 ± 0.65 | 8.89 ± 1.12 |
| AUC$_{0-24}$ (ng·h/mL) | 1575.10 ± 52.91 | 2706.81 ± 119.84 | 2817.83 ± 68.00 |
| AUC$_{0-\infty}$ (ng·h/mL) | 1617.56 ± 62.35 | 3033.50 ± 158.16 | 3149.27 ± 96.01 |
| MRT (h) | 4.19 ± 0.48 | 6.90 ± 0.43 | 7.10 ± 0.59 |
| Relative bioavailability | - | 171.85 | 178.90 |

A comparison between the pharmacokinetic parameters in the three studied groups was conducted using the Tukey-Krammer multiple comparison test ($p < 0.05$). The test showed a statistically significant difference between the commercial tablet and nasal gel (F1) and between the commercial tablet and nasal gel (F17) (P1, P2 < 0.05) regarding C$_{max}$, K$_e$, t$_{1/2}$, AUC$_{0-24}$, AUC$_{24-inf}$, AUC$_{0-inf}$, AUMC$_{24-inf}$, AUMC$_{0-inf}$, MRT, V$_d$, and C$_{ma}$/AUC$_{0-24}$, while there was no statistically significant difference between nasal gel (F1) and nasal gel (F17) (P3 > 0.05). An increase in C$_{max}$, T$_{max}$, AUC$_{0-24}$, and AUC$_{0-\infty}$ of F1 and F17 in comparison with the commercial tablet indicates that the amount of drug absorbed from F1 and F17 via nasal mucosa was significantly higher than that absorbed after oral administration of the commercial product. This indicates improvement in the bioavailability of ITO HCl via the nasal route of administration.

Regarding T$_{max}$, there was a statistically significant difference between the commercial tablet and nasal gel (F1) and between the commercial tablet and nasal gel (F17) (P1, P2 < 0.05), while there was no statistically significant difference between nasal gel (F1) and nasal gel (F17) (P3 > 0.05). Assessed products gave higher T$_{max}$ as compared to commercial tablets due to the presence of mucoadhesive polymer in nasal gel formulations, which increases the contact time and need for the drug diffusion from the polymer. MRT showed a higher value in tested products than the commercial one, indicating the presence of the drug for a longer time in the body after the administration of tested products. There was a statistically significant difference between the three groups regarding AUMC$_{0-24}$ (P1, P2, P3 < 0.05), while there was no statistically significant difference between the three groups regarding TCR (P1, P2, P3 > 0.05).

These results from the Tukey–Kramer multiple comparison test of the pharmacokinetic parameters are represented in Table 4 and Figure 2.

**Table 4.** ANOVA analysis of the pharmacokinetic parameters of the 3 studied groups (commercial tablet, F1, and F17) and a comparison between the 3 groups using Tukey–Kramer multiple comparison test.

| Parameter | Formulations | | | ANOVA *p* Value | P1 | P2 | P3 |
|---|---|---|---|---|---|---|---|
| | Commercial Tablets | F1 | F17 | | | | |
| $T_{max}$ (h) | $0.83 \pm 0.13$ | $2 \pm 0$ | $2 \pm 0$ | 490.00, 0.001 * | 0.001 * | 0.001 * | 0.275 |
| $C_{max}$ (ng/mL) | $608.16 \pm 28.10$ | $673.87 \pm 18.86$ | $664.79 \pm 14.72$ | 16.754, 0.013 * | 0.011 * | 0.035 * | 0.472 |
| $K_e$ ($h^{-1}$) | $0.1655 \pm 0.020$ | $0.0724 \pm 0.0049$ | $0.07906 \pm 0.0103$ | 91.042, 0.001 * | 0.001 * | 0.001 * | 0.398 |
| $t_{1/2}$ (h) | $4.23 \pm 0.47$ | $9.61 \pm 0.65$ | $8.89 \pm 1.12$ | 80.640, 0.001 * | 0.001 * | 0.002 * | 0.134 |
| $AUC_{0-24}$ (ng·h/mL) | $1575.10 \pm 52.91$ | $2706.81 \pm 119.84$ | $2817.83 \pm 68.00$ | 390.735, 0.001 * | 0.002 * | 0.004 * | 0.039 |
| $AUC_{24-inf}$ (ng·h/mL) | $42.46 \pm 15.71$ | $326.69 \pm 57.25$ | $331.43 \pm 88.87$ | 43.160, 0.001 * | 0.001 * | 0.001 * | 0.896 |
| $AUC_{0-inf}$ (ng·h/mL) | $1617.56 \pm 62.35$ | $3033.50 \pm 158.16$ | $3149.27 \pm 96.01$ | 341.756, 0.001 * | 0.002 * | 0.001 * | 0.096 |
| $AUMC_{0-24}$ (ng·h²/mL) | $5769.78 \pm 642.57$ | $13,125.27 \pm 1137.66$ | $14,456.03 \pm 801.87$ | 167.637, 0.001 * | 0.003 * | 0.001 * | 0.02 * |
| $AUMC_{24-inf}$ (ng·h²/mL) | $1019.05 \pm 376.95$ | $7840.597 \pm 1373.89$ | $7954.39 \pm 2132.86$ | 43.159, 0.001 * | 0.0013 * | 0.005 * | 0.896 |
| $AUMC_{0-inf}$ (ng·h²/mL) | $6788.82 \pm 948.72$ | $20,965.87 \pm 2269.20$ | $22,410.42 \pm 2460.27$ | 110.832, 0.001 * | 0.001 * | 0.001 * | 0.232 |
| MRT (h) | $4.19 \pm 0.48$ | $6.90 \pm 0.43$ | $7.10 \pm 0.59$ | 62.322, 0.001 * | 0.002 * | 0.001 * | 0.486 |
| $V_d$ (L) | $13.72 \pm 4.00$ | $34.26 \pm 4.29$ | $26.40 \pm 5.06$ | 32.234, 0.006 * | 0.001 * | 0.001 * | 0.018 |
| TCR (mL/min) | $36.9376 \pm 7.387$ | $41.1472 \pm 4.232$ | $34.12 \pm 2.79$ | 2.80, 0.092 | 0.360 | 0.179 | 0.13 |
| $C_{max}/AUC_{0-24}$ ($h^{-1}$) | $0.38672 \pm 0.0268$ | $0.2492 \pm 0.0080$ | $0.2361 \pm 0.0098$ | 142.296, 0.001 * | 0.001 * | 0.001 * | 0.260 |
| Relative Bioavailability (%) | - | 171.85 | 178.90 | T = 1.01, 0.211 | | | |

P1 comparison between commercial tablet and nasal gel (F1); two comparisons between commercial tablet and nasal gel (F17); P3 comparison between nasal gel (F1) and nasal gel (F17); * Significant difference.

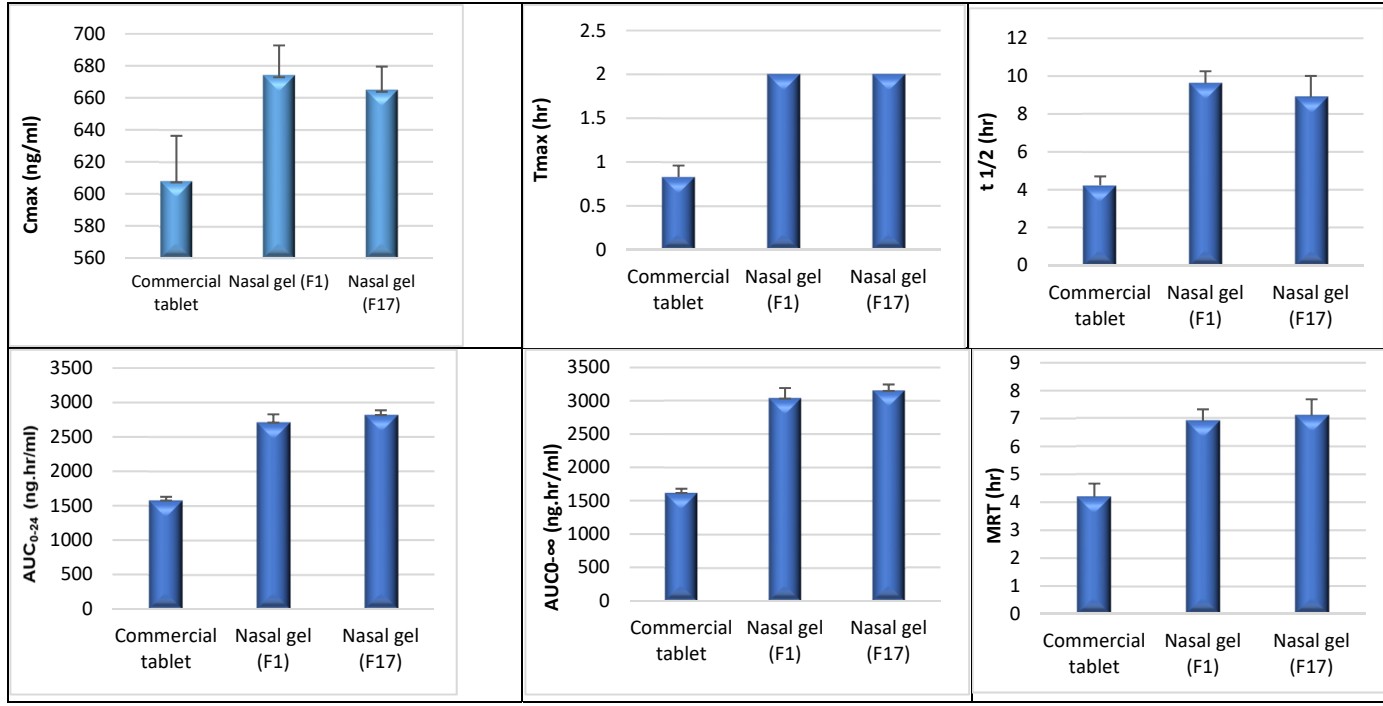

**Figure 2.** A comparison of the pharmacokinetic parameters (Cmax, $T_{max}$, $t_{1/2}$, $AUC_{0-24}$, $AUC_{0-inf}$, MRT) between the three studied groups (commercial tablet, F1, and F17) using a post hoc test.

Relative bioavailability was calculated as a percentage value as follows [22,23]:

$$\text{Relative bioavailability (\%)} = (AUC_{0-24} \text{ test}/AUC_{0-24} \text{ reference}) \times 100, \tag{1}$$

The relative bioavailability of the tested ITO HCl in situ nasal gel (F1) to the commercial tablet based on the mean $AUC_{0-24}$ was found to be 171.85%, and for (F17) to the commercial tablet was 178.90%. This can be seen in Figure 3.

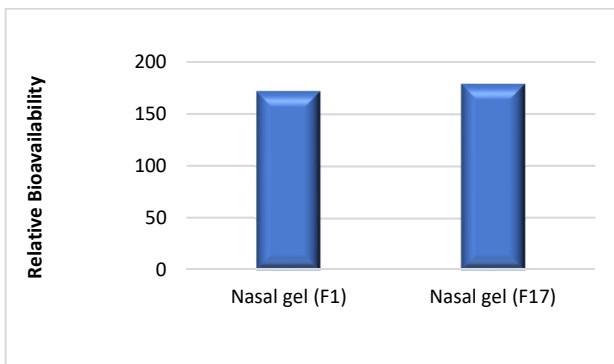

**Figure 3.** Relative bioavailability of in situ gel F1 to the commercial tablet in comparison to the relative bioavailability of F 17 to the same tablets.

The previous results indicate that ITO HCl formulated as in situ nasal gel (F1 and F17) was absorbed into the systemic circulation in a higher amount and remained for a longer duration compared to commercial oral tablets. This was expected due to the avoidance of the first-pass metabolism that was avoided by the nasal administration route, where the drug was directly absorbed into the systemic circulation via nasal mucosa without being metabolized by the liver enzymes.

### 3.2. Histopathological Study

Microscopic examination of control nasal mucosa samples (slide A) shows normal surface epithelium and subepithelial glands of the nasal vestibule, as indicated by the arrows in Figure 4, and an intact surface epithelial lining and cartilaginous layer of the nasal septum as arrows show in Figure 5.

Microscopic examination of the test samples treated with gel (F1 and F17) revealed a normal histological structure without any necrosis, edema, hemorrhage, irritation, or erosion in both nasal vestibule and nasal septum. A comparison of a photomicrograph of control slide (A) with test slides B (F1) and C (F17) shows a normal histological structure of the surface epithelium and subepithelial glands (Figure 4B,C) and intact surface epithelial lining and cartilaginous layer (Figure 5B,C). No residual gel was seen on the H&E-stained slides.

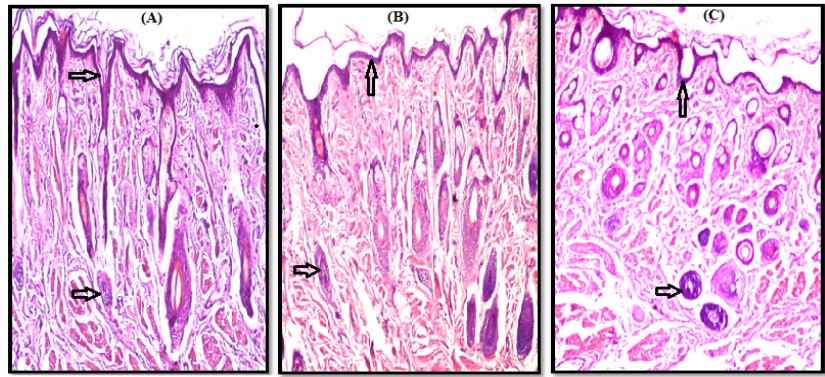

**Figure 4.** Photomicrograph of nasal vestibule showing a normal histological structure of surface epithelium and subepithelial glands as shown by arrows, (**A**) Control, (**B**) F1, and (**C**) F17.

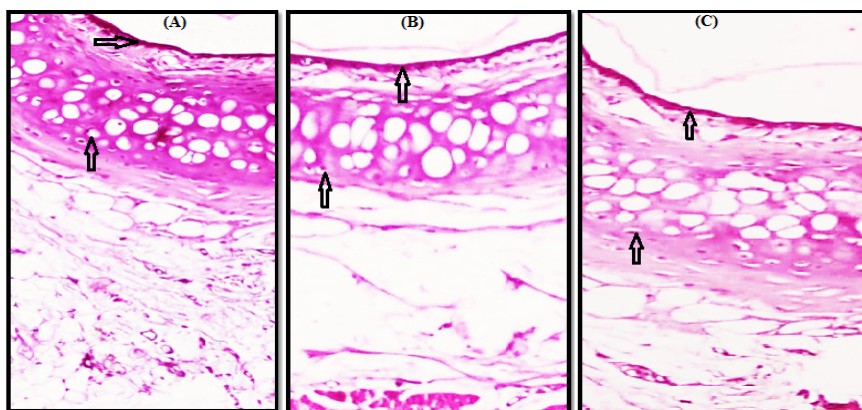

**Figure 5.** Photomicrograph of nasal septum showing intact surface epithelial lining and cartilaginous layer, as shown by arrows indicating the absence of any degenerative changes, (**A**) Control, (**B**) F1 and (**C**) F17.

## 4. Conclusions

The results showed that the percentage relative bioavailability of ITO HCl in situ nasal gel was found to be 171.85% and 178.90%, respectively, compared to the commercial oral tablet. Increasing the bioavailability of ITO HCl from the nasal formulations reflects the potential of ITO HCl nasal formulations to bypass the hepatic metabolism of the drug. Morphological studies revealed the safety of nasal in situ gel F1 and F17 to the nasal mucosa after its application for 14 days in rats. Both F1 and F17 nasal in situ gel formulae showed a promising safe non-invasive route to replace the oral route for the administration of ITO HCl with better bioavailability. Based on the previous study, formulations F1 and F17 are recommended for further clinical studies.

**Author Contributions:** Conceptualization, M.A.M.; investigation and methodology, A.I.A.E.-F.; writing—original draft, review and editing, D.A.O. and R.A.A.; supervision and project administration, M.A.M. All authors have read and agreed to the published version of the manuscript.

**Funding:** This research received no external funding.

**Institutional Review Board Statement:** The study was conducted according to the guidelines of the Guide for the Care and Use of Laboratory Animals, and approved by the ethical committee, Faculty of Pharmacy, Al-Azhar University (permit number: 136, Date of approval: 21 October 2017).

**Data Availability Statement:** The data presented in this study are available in the paper or the here.

**Acknowledgments:** The authors would like to express their deep gratitude to the pharmaceutics and industrial pharmacy department in the Faculty of Pharmacy (Girls), Al-Azhar University for their support. The authors wish to express their appreciation to MUP Pharm for Pharmaceutical Industries (Egypt) for providing ITO HCl, and the Egyptian International Pharmaceutical Company (EIPICO), Egypt for their generous gift of poloxamer 407.

**Conflicts of Interest:** The authors declare no conflict of interest.

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
