# Peer review of "Pharmacokinetic Study of Mucoadhesive Itopride Hydrochloride In Situ Nasal Gel Formulations in a Comparative In Vivo Study and Histopathological Safety Evaluation"

_scipharm, doi:10.3390/scipharm90010008_

Round 1

Reviewer 1 Report

This manuscript is well written and the result presented is promising. I have the following comments:

1. page 3 lines 111-112 stated that there were 6 rabbits in the study and they were equally divided into 3 groups. Does that mean each group had two rabbits? However, page 3 lines 115-116 stated that a single-dose cross-over study design with a washout period was followed. I would like to believe that sample size was 6 for each of the three groups tested, in which case I would have less concern of sample size of this study. Please clarify and give the detail of the cross-over study design.

2. There are a few areas where units are wrong or not clear. Page 5 line 213, 216, 217 stated “microgram” and should be corrected to “nanogram.” Unit for elimination rate constant should be hr-1 not hr-1. Page 4 line 152: please specify 4-6 micrometers. page 5 line 217: change"Keg" to "Ke." In addition, please leave a space between number and unit.

3. page 6 line 234: the first “F17” should be corrected to “F1.”

4. Questions related to study on rabbits and rats. Were 6 rabbits sacrificed after the study? If yes, why histopathological study was not done on rabbits rather than initiating another study on rats? For group II and group III studies done on rabbits, was the gel applied to only one nostril or both nostrils? Page 4 section 2.2.7 stated that 20 microliters of gel were applied to the right nostril of the rats. In the study done on rats how was the dosing related to ITO HCL equivalent to 2.5 mg/kg applied on rabbits? Why the gels were applied 14 days daily, instead of only one day or daily for a few other days?

5. For the histopathological study done on rats, how many rats were used? How many slides were examined for each rat? Any attempt to address the issue that any adverse effect of applying nasal gel could be focal and easily missed while multi-level examination of the paraffin-embedded block might need be performed? Was any residual gel seen on the H&E stained slides? A picture showing residual gel would be more convincing. Please emphasize that there was no erosion, ulceration, or inflammation identified on the nasal sample from rats with nasal gel application.

6. page 1. line 38: change "GIT" to "gastrointestinal tract (GIT)" 

page 8 line 274, 278, 282, 283. Fill in with “Fig. 4” or “Fig. 5” when appropriate.

Author Response

Dear Sir,

Kindly find attached a file of response.

Reviewer 2 Report

In an earlier study, the authors investigated the formulation of an in situ nasal gel containing itroprid. This manuscript is a continuation of this research work. The main pharmacokinetic parameters of the two formulations and a commercially available formulation containing itropide were determined by in vivo animal experiments. The results were compared statistically. Although the manuscript is interesting, it needs to be improved in several places. 
- The batch size during gel preparation is not indicated. 
- Several equipment types, manufacturer missing (ultrasonic equipment, vortex, vacuum oven) 
- In Table 2 standard deviations of the Tmax values are missing, but all the data can be found in Table 3, so it is unnecessary to convey the second table separately. 
-In line 260: the end of a sentence in a line is not visible 
-In 3.2. chapter four times error can be read. 
-What do the arrows in Figures 4 and 5 mean? 
-The chapter of Fig. 5 needs to be improved 
- The english language of the manuscript needs to be checked again. 

Author Response

Dear Sir,

Kindly find attached a file of response to the comments.

Best regards.

Round 2

Reviewer 2 Report

Thank you for your answer. I have no further questions.